# Nano-Based Drug Delivery Systems for Periodontal Tissue Regeneration

**DOI:** 10.3390/pharmaceutics14102250

**Published:** 2022-10-21

**Authors:** Huanhuan Chen, Yunfan Zhang, Tingting Yu, Guangying Song, Tianmin Xu, Tianyi Xin, Yifan Lin, Bing Han

**Affiliations:** 1Department of Orthodontics, School and Hospital of Stomatology, Peking University, Beijing 100081, China; 2National Engineering Laboratory for Digital and Material Technology of Stomatology, Beijing Key Laboratory of Digital Stomatology, Beijing 100081, China; 3Division of Paediatric Dentistry and Orthodontics, Faculty of Dentistry, The University of Hong Kong, Hong Kong, China

**Keywords:** periodontitis, tissue regeneration, nano-based drug delivery systems

## Abstract

Periodontitis is a dysbiotic biofilm-induced and host-mediated inflammatory disease of tooth supporting tissues that leads to progressive destruction of periodontal ligament and alveolar bone, thereby resulting in gingival recession, deep periodontal pockets, tooth mobility and exfoliation, and aesthetically and functionally compromised dentition. Due to the improved biopharmaceutical and pharmacokinetic properties and targeted and controlled drug release, nano-based drug delivery systems have emerged as a promising strategy for the treatment of periodontal defects, allowing for increased efficacy and safety in controlling local inflammation, establishing a regenerative microenvironment, and regaining bone and attachments. This review provides an overview of nano-based drug delivery systems and illustrates their practical applications, future prospects, and limitations in the field of periodontal tissue regeneration.

## 1. Introduction

Periodontitis is a multifactorial non-communicable inflammatory disease characterized by chronic progressive destruction of the tooth-supporting apparatus, affecting all parts of the periodontium, and causing irreversible damage [1]. Aiming to obviate inflammation of the periodontium and arrest tissue destruction, initial regimens were mostly directed toward dental plaque and calculus to reduce periodontal pathogens by mechanical scaling and root planning, and occasionally, adjuvant therapy with antiseptics or antibiotics [2]. Moreover, reconstructing the defected periodontal tissue and rehabilitating the functional and esthetic properties of the stomatognathic system are ultimate goals [3]. As naked growth factors or other molecules could be quickly deactivated or easily diffused and uncontrolled drug release manners would not meet the physiological need, which may lead to undesirable even adverse outcomes such as dysbiosis and tumorigenesis, more advanced drug delivery systems are demanded in periodontal regenerative medicine [4,5]. With the notable progression of nanotechnology, various organic or inorganic nanoplatforms have been exploited as candidates for nano-based drug delivery systems (nano-DDSs) [6]. Nano-DDSs are forms of submicron particles, tubes, micelles, hydrogels, etc., commonly 10–1000 nm in diameter [7]. It is acknowledged that tissue repair and regeneration feature interdependent and/or connected biological events controlled by a large variety of biomolecules providing feedback at the damaged sites and controlling the mechanisms beyond the healing [8]. Hence, it is not surprising that nano-DDSs can play an important role in overcoming the existing challenges in tissue regeneration, given their capability for displaying material properties at the cellular and molecular level [9]. Nano-DDSs have unique advantages: first, benefited from large specific surface-to-volume ratio, nano-DDSs possess high drug loading capacities. Nanoscale carriers also have preponderances in promoting solubility and permeability of hydrophobic drugs and prolonging the half-life of drugs in vivo, etc. In addition, the surface of nanoparticles can be simply modified, endowing nano-DDSs with targeting transport and environmentally responsive release performances, to achieve precise treatment. Owing to these features, nano-DDSs have become a focus in cancer therapy, gene therapy and regenerative medicine [10].

As periodontal regenerative medicine demands targeted and controlled delivery of diverse biomolecules that can promote cell adhesion, proliferation, differentiation and tissue formation, the optimization of nano-DDSs can be of significant help for eventually repairing or replacing damaged periodontal tissues. Herein, this review aims to give a brief overview of periodontitis and its pathogenesis, clarify the available targets in periodontal anti-inflammation and regeneration, and report different nano-DDSs for periodontal tissue regeneration, including significant applications of natural and synthetic compound-based nanomedicines and challenges associated with nanomaterials in medicines.

## 2. Pathogenesis of Periodontitis

Periodontitis is a progressive inflammatory disease interrelated with the accumulation of dental biofilm and bacteria-host immune system-mediated dysbiosis [1]. Its pathogenesis can undergo various stages, from gingivitis evolving into periodontitis [11]. The common characteristics of periodontitis include periodontal tissue inflammation, bleeding upon probing, periodontal attachment loss, deep probing pockets, radiographic evidence of alveolar bone resorption, mobility of supporting teeth, and pathologic migration. The etiology of periodontitis is multifactorial, mainly attributing to complex dysbiosis among periodontal pathogens, host inflammatory and immune responses, and other environmental factors. Some systemic promoting factors such as genetic factors, endocrine disorders, malnutrition, immunodeficiency, smoking, mental stress, etc. can reduce the host’s defense ability or aggravate the inflammatory reaction of periodontal tissue [12].

The invasion of specific periodontal pathogens, such as *Tannerella forsythia*, *Porphyromonas gingivalis*, *Treponema denticola*, etc., and their products, such as lipopolysaccharide (LPS) along with other virulence factors, initiate a host inflammatory and immune response, in which individual variability in the host’s response to local factors may play a significant role in prognosis of periodontitis [13]; for example, the destructive host immune responses are more likely to occur in a susceptible host [14]. The host inflammatory and immune responses recruit massive neutrophils, macrophages, dendritic cells, and lymphocytes, and then produce a variety of pro-inflammatory cytokines, including interleukin (IL)-1β, tumor necrosis factor (TNF)-α, and prostaglandin E2 (PGE2). These resulting pro-inflammatory cytokines along with other virulence factors stimulate the expression of matrix metalloproteinases (MMPs) by neutrophils, fibroblasts, macrophages, and junctional epithelial cells. The presence of MMPs then lead to the destruction of collagen fibers. In addition, the differentiation and maturation of osteoclasts are activated through the interaction between receptor activator of nuclear factor κB (NF/κB) (RANK) on osteoclast precursors and RANK ligand (RANKL) on the T helper cells and osteoblasts. The mature osteoclasts then mediate the destruction of alveolar bone. Without therapeutic interventions, the degradation products of inflammatory and immune responses become nutrients for pathogenic bacteria, further increasing the imbalance of subgingival dental biofilm and promoting graver tissue-destructive inflammation. Consequently, instead of killing pathogenic bacteria and eliminating pathogenic factors, this extensive immune response of the susceptible host destroys the patient’s own tissue (Figure 1) [15].

## 3. Available Targets in Periodontal Anti-Inflammation and Regeneration

The therapeutic purpose of periodontal tissue regeneration is to concurrently eliminate inflammation and reconstruct the function and structure of periodontal tissues. Because nanostructures display the advantages of giving biochemical reactions at cellular, subcellular or molecular levels, numerous biomaterials in the nanoscale range are adopted to deliver therapeutic drugs to specific targeted sites in a controlled manner. Then it is important to clarify the available targets in periodontal anti-inflammation and regeneration [16], to achieve accurate targeted treatment (Table 1).

### 3.1. Regulator of G-Protein Signaling (RGS)

Osteoclasts, derived from mononuclear hematopoietic myeloid lineage cells, are multinucleated cells that degrade bone matrix. The mechanism of regulating osteoclast formation and their functions are complex, while calcium is one important signal for osteoclast motility downstream of tyrosine kinase signals. There is growing recognition that regulators of G-protein signaling play important roles in regulating calcium oscillations and thus osteoclast differentiation. More than 30 RGS proteins have been found at present, among which RGS10, the smallest protein in the RGS family [17], and RGS12, the largest protein in the RGS family [18], were prominently expressed in osteoclasts. Numerous studies have indicated that both RGS10 and RGS12 play roles in regulating calcium oscillations and osteoclast differentiation through RANKL-evoked calcium oscillation-nuclear factor of activated T cells (NFAT) c1 signaling pathway (Figure 2) [19]. In addition, Yuan et al. have found that the deletion of RGS12 inhibits osteoclastogenesis, as well as the expression of inflammatory cytokines and macrophages migration in response to the stimulation of lipopolysaccharide (LPS), which demonstrates that the restraint of RGS12 in macrophages is a potential therapeutic target for preventing bone loss in periodontitis treatment [20].

### 3.2. Mitogen-Activated Protein Kinase (MAPK) Signaling Pathway

Mitogen-activated protein kinases (MAPKs), a class of serine/threonine protein kinases and signal transduction mediators, play an important role in the regulation of inflammation. Activation of the p38MAPK signaling pathway may be involved in reducing lipopolysaccharide (LPS)-induced bone resorption in periodontitis. Investigations have showed that several molecules can regulate osteogenic differentiation of periodontal ligament stem cells (PDLSCs) via p38 MAPK signaling pathway. For example, cerebellar degeneration-related protein 1 transcript (CDR1as), a newly discovered circular RNA (circRNA), has been reported to be an miR-7 inhibitor, triggering the upregulation of growth differentiation factor 5 (GDF5) and subsequent Smad1/5/8 and p38 MAPK phosphorylation to induce osteogenic differentiation of PDLSCs [21]. The LL-37, a human antimicrobial peptide (AMP), has various biological functions and potentials, inhibiting inflammation and promoting bone marrow stromal cell (BMSC) osteogenesis via P2X7 receptor (P2X7R) and MAPK signaling pathway [22]. Mineral trioxide aggregate (MTA), a bioactive material, can promote the odonto/osteogenic capacity of hPDLSCs via activating the NF-κB and MAPK pathways (Figure 3) [23].

### 3.3. The NF-κB Signaling

The prototypical proinflammatory signaling pathway NF-κB is essential for expression of proinflammatory genes including cytokines, chemokines, and adhesion molecules. Previous research has established that NF-κB signaling mediates RANKL-induced osteoclastogenesis, whose inhibition is effective to restrain osteoclast formation and bone resorption [24]. The NF-κB activator 1 (Act1), mainly expressed in immune cells, modulates immune cells’ function to regulate inflammation in periodontitis. It is possible that downregulation of macrophage-specific Act1 could aggravate periodontitis and accelerate alveolar bone loss via TNF/NF-κB signaling [25]. Potassium dihydrogen phosphate (KH2PO4), a kind of inorganic phosphate added to the solution, can improve the proliferation and odonto/osteogenic differentiation capacity of the PDLSCs via NF-κB pathway, and thus serves as a potential target for periodontal regeneration in clinical treatments [26]. Mineral trioxide aggregate (MTA), a biocompatible material, can enhance the odonto/osteogenic capacity of inflammatory dental pulp stem cells (iDPSCs) by activating the NF-κB pathways [27].

### 3.4. The Wnt Signaling

The Wnt signaling is significantly involved in the development and homeostasis of tissues via regulation of their endogenous stem cells. Investigations have shown that Wnt signaling is important for homeostasis of periodontal tissues, from functional periodontal cells to progression of periodontitis [28]. As such, biomolecules targeting Wnt signaling may be an optimal therapy for periodontitis treatment. Sclerostin and DKK1 act as the Wnt signaling inhibitors and are upregulated in patients with chronic periodontitis. Targeting sclerostin and DKK1 could be an attractive adjuvant strategy for periodontitis treatment [29]. Berberine, a benzylisoquinoline plant alkaloid from Coptidis Rhizoma, shows antibacterial actions for *Porphyromonas gingivalis* and promotes osteogenic differentiation via Wnt/β-catenin signaling pathway, which represent as a prospective drug for periodontal tissue regeneration [30]. Baicalein [31] and Parthenolide [32] also have positive effects on Wnt signaling, which enable their possible use for treatment of periodontitis. Romosozumab, a monoclonal antibody against sclerostin, is considered as a therapeutic agent that targets Wnt signaling [33] for osteoporosis and fragility fractures treatments [34]. As periodontitis and osteoporosis have homologous mechanisms, and osteoporosis is recognized as a hazard factor for periodontal defects and tooth loss, systemic romosozumab administration can promote periodontal regeneration as local delivery cannot repair the alveolar bone defects. Lipopolysaccharide (LPS), the major element of the outer membrane of gram-negative bacteria, is a pertinent harmful factor in the oral microenvironment. Xing et al. have found that *Escherichia coli*-derived LPS can stimulate proliferation and osteogenic differentiation of PDLSCs under the activation of TAZ (transcriptional activator with a PDZ motif), while the increment and activation of TAZ were mostly mediated by the Wnt/β-catenin pathway [35]. In addition, erythropoietin (EPO) [36], estrogen [37], kaempferol (a type of flavonoid) [38], and vitamin K2 (menaquinone 4, MK-4) [39] all have been reported to promotes osteogenic differentiation of the PDLSCs via Wnt/β-catenin signaling pathway, which suggests the potential application for periodontal tissue regeneration.

**Table 1 pharmaceutics-14-02250-t001:** Available targets in periodontal anti-inflammation and regeneration.

Targeting Pathway	Target Cell	Regulatory Mechanism	Regulating Molecules	Reference
Regulator of G-protein signaling (RGS)	Osteoclasts	regulating calcium oscillations and thus osteoclast differentiation	RGS10, the smallest protein in the RGS family	Almutairi et al. [17]
			RGS12, a multi-domain and the largest protein in the RGS family	Yang et al. [18]
Mitogen-activated protein kinase (MAPK) signaling pathway	Periodontal ligament stem cells (PDLSCs)	reducing inflammatory cytokine biosynthesis and bone resorption, regulating osteoblastic differentiation of PDLSCs	Cerebellar degeneration-related protein 1 transcript (CDR1as), a newly discovered circular RNA (circRNA)	Li et al. [21]
			LL-37, a human antimicrobial peptide (AMP)	Yu et al. [22]
			Mineral trioxide aggregate (MTA), a bioactive material	Wang et al. [23]
The NF-κB signaling	Osteoclast, Macrophages	Inhibiting osteoclast formation and bone resorptive activity, regulating inflammation	NF-κB activator 1 (Act1), mainly expressed in immune cells	Pathak et al. [25]
			Potassium dihydrogen phosphate (KH2PO4), a kind of inorganic phosphate into the solution	Xu et al. [26]
			Mineral trioxide aggregate (MTA), a biocompatible material	Wang et al. [27]
The Wnt signaling	Periodontal ligament stem cells (PDLSCs)	Regulating growth, development, and homeostasis of the organism, controlling cell fate such as proliferation, differentiation, canceration, and apoptosis	Sclerostin and DKK1, Wnt signaling inhibitors	Witcher et al. [29]
			Berberine, a benzylisoquinoline plant alkaloid from Coptidis Rhizoma	Zhang et al. [30]
			Baicalein, an active ingredient extracted from the traditional Chinese herb *Scutellaria baicalensis* Georgi	Chen et al. [31]
			Parthenolide (PTL), an active constituent of the plant *Tanacetum parthenium*	Zhang et al. [32]
			Romosozumab, a monoclonal antibody against sclerostin	Paik et al. [33] & Ishibashi et al. [34]
			*Escherichia coli*-derived Lipopolysaccharide (LPS)	Xing et al. [35]
			Erythropoietin (EPO), a glycoprotein cytokine	Zheng et al. [36]
			Estrogen, 17β-estradiol	Jiang et al. [37]
			Kaempferol, a type of flavonoid	Nie et al. [38]
			Vitamin K2, a product of intestinal bacterial metabolism in the body; menaquinone 4 (MK-4), one of the most active members among the vitamin K2 family	Cui et al. [39]

## 4. Nano-Drug Delivery Systems

Numerous novel nano-DDSs have been designed for periodontitis treatment for advantaged characteristics, such as pharmacodynamics, bioavailability, and cell-specific targeting [40]. The selection of appropriate nano-DDSs for specific substances should be in harmony with the physiology of the deteriorated tissue and the pharmacology of the agents needed. For example, liposomes, dendrimers, or hydrogels, can provide a sustained, controlled release in the damage site for proteins or polypeptides. In addition, it will be advantageous if the selected drugs or drug delivery systems can be easily modified according to the desired properties of the targeting site [41]. The following section describes two main forms of nano-DDSs that have been researched sufficiently for periodontal tissue regeneration [42]: nanoparticles and nanofibers (Figure 4).

### 4.1. Nanoparticulate Delivery Systems

Nanoparticles enhance the in vivo efficacy of bioactive molecules for easy penetration, better drug-release kinetics, and controlled delivery required for successful periodontal tissue regeneration [3]. There are multiply kinds of nanoparticulate delivery systems [43], including liposomes, solid lipid nanoparticles, polymeric nanoparticles, inorganic nanoparticles, nanotubes, dendrimers, and micelles, whereas nanoparticulate delivery systems for periodontal tissue regeneration that have been already investigated include liposomes, polymeric nanoparticles, inorganic nanoparticles and nanocrystals, and dendrimers (Table 2) [44].

#### 4.1.1. Liposomes

Liposomes are well recognized as effective drug delivery systems for the bi-layered structural versatility, biodegradability, biocompatibility, non-toxic and non-immunogenicity nature [45]. Liposomes possess the unique ability to deliver molecules with different solubility as structurally, they have the polar head groups oriented to the inner and outer aqueous phase (Figure 5) [46]. Several variables such as the drug/lipid ratio, drug releasing kinetics and retention, encapsulation efficiency, cost efficiency and liposome stability should be considered when selecting an appropriate method for drug to be encapsulated into liposomes [47]. The physical action of ultrasound has been reported to be able to deliver therapeutic biomolecules for periodontal regeneration. After successfully developed a useful carrier “Bubble liposomes” for gene or drug delivery, Suganos et al. then examined the probability of delivering genes into gingival tissues combining Bubble liposomes with ultrasound. Fortunately, the combination of Bubble liposomes and ultrasound offered an efficient strategy for delivering plasmid DNA into the gingiva [48], which may help the treatment of periodontitis [49]. In addition, negatively charged liposomes are considered as a multipurpose tool in the field of drug delivery systems for their hydrophobic and hydrophilic characteristic [50]. Moreover, liposomes modified with viral fusion proteins are also capable of introducing encapsulated antigenic substances into cell cytosol [51]. Hu et al. have synthesized a novel pH-activated nanoparticle comprising a quaternary ammonium chitosan, i.e., N,N,N-trimethyl chitosan, a liposome, and doxycycline (TMC-Lip-DOX NPs). It was found that the TMC-Lip-DOX NPs could achieve excellent inhibition of free mixed bacteria and biofilm formation, and show superb biocompatibility with hPDLSCs, which indicated a good potential use for treatment of periodontal inflammation [52].

Several studies have found that macrophages could coordinate inflammation resolution and healing by restraining proinflammatory stimuli, eliminating dead cells through efferocytosis, and improving neovascularization required for tissue regeneration, so it is important to focus on the macrophage activation states in bone regeneration [53]. As the structure and function of liposomes closely resemble the biological environment, the deposition of lipid components at specific sites leads to the drug load acting on the macrophage. That is why the 2% minocycline hydrochloride liposomes showed significantly more effective inhibition of TNF-α secretion in macrophages compared with periocline and minocycline hydrochloride solution, which is important to attenuate the destructive impact of cytokine-mediated tissue damages in periodontitis. Because liposomes display viable targets for natural macrophage phagocytosis, Liu et al. prepared minocycline hydrochloride liposomes that could be delivered to sites sufficient in macrophages [54]. Resveratrol (RSV), a nonflavonoid polyphenol, is known for its immunomodulatory and anti-inflammatory characters. Shi et al. have developed a novel resveratrol-loaded liposomal system (Lipo-RSV) to inhibit inflammatory progression and found that Lipo-RSV was biocompatible and could transform the inflammatory macrophages from M1- to M2-like phenotype through downregulating p-STAT1 and activating p-STAT3. The pro-inflammatory cytokines, such as IL-1β, IL-6, and TNF-α, could be reduced by Lipo-RSV scavenging reactive oxygen radicals (ROS) and inhibiting the NF-κB signal and inflammasomes, which revealed that Lipo-RSV could be a potential drug delivery system for periodontal diseases without antibiotic treatment [55].

Despite the general success in drug delivery, there are also several difficulties for liposomes being faced, such as limited chemical and physical stability, drug leakage, and difficulties in augmenting the output procedure; however, it is fortunate that some disadvantages of liposomes can be overcome, and additional properties can be added by modifying the vesicular composition and features [56]. Then the creation of the so-called stealth, long circulating, or PEGylated liposomes were developed. The stealth strategy mainly means to modify the liposomal membrane surface with biocompatible hydrophilic polymer conjugates, such as polyethylene glycol (PEG), chitosan, etc., thereby increasing repulsive forces between liposomes and serum-components, reducing immunogenicity, enhancing the blood circulation half-life, avoiding macrophage uptake, and reducing the toxicity of encapsulated compound [57].

#### 4.1.2. Polymeric Nanoparticles

Polymers play a crucial role in drug delivery systems due to their favorable biochemical and physicochemical properties. The characteristics such as non-immunogenicity, biological inactivity, and the facility of functional groups for covalent coupling of target moieties are what distinguishes them from inorganic nanoparticles. A large variety of natural and synthetic originated polymeric nanoparticles-carriers have been used for the treatment of periodontal defects. Furthermore, different methods have been adopted to improve the therapeutic effect of polymeric nano-DDSs, such as tailoring the structures, combined using biodegradable polymers, or modifying the carrier surface to achieve prolonged retention at the damaged sites [58]. The following section will introduce several commonly used polymeric nanoparticles applied in periodontal tissue regeneration.

Chitosan (CHT), a deacetylated derivative of chitin, is one of the most often investigated as a nanometric excipient for treatment of periodontal defects due to its advantageous properties, such as antibacterial activity, biodegradability, biocompatibility, non-toxicity, tissue healing, and osteoinducting effects [59], which enable it to effectively promote cell adhesion, proliferation, and differentiation [60]. Chitosan can be a flexible material in hydrated environments, which is advantageous over other rigid synthetic materials like polyglycolic acid (PGA) and polylactic acid (PLA). Due to its biodegradability profile that it can be retained as a barrier for up to 4–6 weeks, chitosan has been used to provide positive reconstruction of periodontal tissue [61]. The biodegradation of chitosan membrane prepared by different techniques have been reported and about 15–40% degradation of pure chitosan membrane was reported after 90 days in phosphate buffer saline (PBS) at normal temperature [62]. In another investigation, the tailoring of degradation and release rates of chitosan can be set for a specific period in bone tissue regeneration [63]. In addition, chitosan also displays a cationic character and can form stable complexes with relatively large polyanionic molecules such as nucleotides, which represent a possible drug carrier for non-viral gene delivery [64]. However, although chitosan may serve as an effective templet for osseous defects repair, its osteo-conductivity needs to be improved for ideal bone regeneration. As such, it is necessary to combine chitosan with other bioactive materials to improve the bioactivity. Recently, composite biomaterials based on biodegradable polymers containing bioactive glasses have been reported to show excellent bioactivity, biocompatibility, and osteo-conductivity both in vitro and in vivo [65]. Mota et al. have proposed a combination of chitosan (CHT) with bioactive glass nanoparticles (BG-NPs) for guided bone regeneration and found that the addition of BG-NPs to CHT membranes decreased the mechanical properties but improved the bioactivity, such as inducing the sediment of bone-like apatite in simulated body fluid (SBF) and promoting cell metabolic activity and mineralization [66]. Moreover, chitosan/plasmid nanoparticles encapsulating platelet-derived growth factor (PDGF) has been reported to be incorporated into a porous chitosan/collagen composite to achieve steady and sustained releasing of growth factors over 6 weeks, which offered a 3D carrier for increased proliferation of periodontal ligament cells (PDLSCs) [67].

**Figure 5 pharmaceutics-14-02250-f005:**
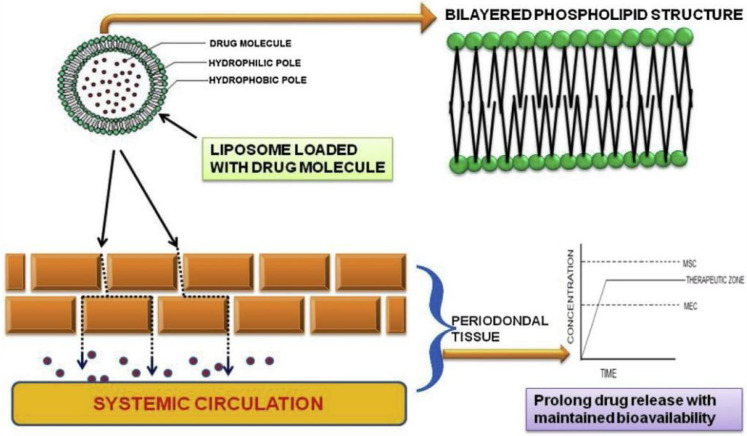
Schematic diagram of bi-layered structure of liposome with penetration into the periodontal tissue. Reprinted/adapted with permission from Ref. [62]. 2019, Suresh, P.K.

Nanogels, nanosized crosslinked hydrophilic polymeric networks, have been widely explored for drug delivery due to their unique properties, such as biocompatibility, stability, tunable particle size, large capacity of drug loading, and capacity of surface modification for effective and specific targeting [68]. The dimensions of nanogels are less than 200 nm in diameter, making themselves suitable carriers for the delivery of therapeutic drugs, siRNAs, and peptides [69]. Nanogels can also be added to modify the inner texture, architecture, and mechanical properties of tissue scaffolds to regulate cell behavior [70]. In addition, nanogels can prevent aggregation or denaturation of proteins [71]; for example, higher stability of proteins encapsulated within the crosslinked nanogels was reported even at temperatures higher than physiological values and in the existence of organic solvents [72]. Alles et al. have used the W9-peptide, a TNF-α and RANKL antagonist, as a model to examine the feasibility of cholesterol-bearing pullulan (CHP)-nanogel as the drug delivery system. It was found that CHP-nanogel served as a fine carrier for the W9-peptide and prevented its aggregation, which revealed the feasibility of CHP-nanogel-mediated peptide delivery in preventing bone resorption in vivo [73]. In addition, He et al. have fabricated a new asymmetric barrier membrane to address the challenges of limited osteoconductive and antibacterial potential of currently available membranes. With nanoscale agarose hydrogel functioning as the main body of the barrier membrane, hollow carbonated hydroxyapatite (CHA) being sedimented in agarose to exhibit an asymmetrical structure, and ε-poly-lysine (ε-PLL) being selected as an antimicrobial agent to equip the membrane with long-lasting antibacterial activity, the barrier membrane has showed higher mechanical properties and better biocompatibility, indicating its potential for periodontal tissue engineering [74].

Poly(lactic-co-glycolic acid) (PLGA), a synthetic biodegradable polymer, has been widely used as a carrier for therapeutic drug delivery systems due to its biodegradability, biocompatibility, suitable biodegradation kinetics and tractable mechanical properties. These properties and versatility allow PLGA to encapsulate and deliver a large variety of hydrophobic and hydrophilic molecules. A controlled and cell-targeted delivery of BMP2 using PLGA nanoparticles as a main component have been examined, in which the targeting properties was supplied by conjugating a cell-specific ligand to direct the release of encapsulated therapeutic drugs preferably in close interaction with the target cells [75]. In addition, a study aiming to produce a blend of poly(lactic-co-glycolic acid)/chitosan/Ag nanoparticles for periodontal regeneration and to investigate the optimal composite ratio of these three materials was conducted by Xue et al. It was found that the nPLGA/nCS/nAg complex was non- cytotoxic and contributed to cell mineralization, and the 3:7 ratio of nPLGA/nCS and 50 µg/mL nAg were found to be the optimal proportion of the three materials [76].

Polytetrafluoroethylene (PTFE), a stable polymer both chemically and biologically inert [77], is commonly used for GTR membranes because it can resist microbiological and enzymatic attacks and its porous microstructure allows connective tissue in growth. Expanded polytetrafluoroethylene (e-PTFE) membranes have also been applied widely in periodontal regeneration [78]. The e-PTFE membranes and other resorbable membranes usually require primary suture or soft tissue coverage to prevent the ingrowth of soft tissue, membrane migration and degradation, and graft exposure. Moreover, an early bacterial infection can also occur in e-PTFE, which would affect the curative effect of tissue regeneration. Next, high-density polytetrafluoroethylene membranes (n-PTFE) emerged as a substitute to e-PTFE [79]. The n-PTFE, made of 100% absolute bio-inert PTFE, is highly dense, non-expanded, non-porous, and non-permeable. Available literature related to n-PTFE membranes have proved its efficacy in guided tissue regeneration (GTR) and guided bone regeneration (GBR). In addition, a recent study on designing antimicrobial coatings for PTFE membranes has been achieved by applying a mussel-inspired approach and in situ formation of silver nanoparticles (AgNPs), which provides a novel versatile approach for AgNPs coating for many other types of membranes through applying the adhesive behavior of mussel inspired coatings [80].

Polycaprolactone (PCL) membrane synthesized by electrospinning technique can mimic the extracellular matrix (ECM), combining both core-shell and nano-reservoirs functionalization [81]. To promote specific regeneration of the damaged tissues, it is of great use to combine a passive release of anti-inflammatory drugs with nano-reservoir containing pro-regenerative substances. Strub et al. have researched the kinetics of maxillary bone regeneration in a pre-clinical mouse model of jawbone lesion treated with BMP-2/Ibuprofen encapsuled PCL membranes [82] and found that PCL functionalized biomembrane promoted bone regeneration. The mechanism may be that the passive release of ibuprofen will reduce the inflammation leading to increased BMP-2 secretion by macrophages, while active loading of BMP-2 will directly promote the regeneration of targeted tissues [83], which provides a strategy that bringing into other osteogenic, osteoinductive, and angiogenic molecules to promote osteogenesis and neovascularization of tissue-engineered bone.

Chorion membrane (CM) and amnion/chorion membrane (ACM) have emerged recently as the optimal choice for the GTR membrane. In addition to biocompatibility, low immunogenicity, permeability, stability, and resorbability, they also possess antifibrotic, anti-inflammatory, antimutagenic characters, and pain-relieving effects [84]. Several researchers have found that CM and ACM could be applied for treatment of intra-bone and furcation defect, gingival recession, alveolar ridge preservation, maxillary sinus membrane repair, and large bone defect reconstruction, but further studies are needed to provide experimental evidence and particularly to demonstrate their role in tissue regeneration [85].

#### 4.1.3. Inorganic Nanoparticles and Nanocrystals

The most commonly investigated inorganic nanoparticles for the treatment of periodontitis are metallic nanoparticles with antimicrobial and regenerative features and calcium-containing nano-biomaterials for bone regeneration [86]. Compared with organic nanoparticles, inorganic nanoparticles have several advantages such as thermal resistance, chemical stability, and long-lasting action.

Strontium (Sr^2+^) is a cation that can stimulate the differentiation of mesenchymal stem cells (MSCs) to form bone tissue by inhibiting the activity of osteoclasts. Strontium ranelate has been recognized as a therapeutic drug for the treatment of osteoporosis, as well as for bone reconstruction, which presents concomitant osteoanabolic and anti-resorptive dual biological activity [87]. There were recent studies demonstrating that strontium treatment significantly increased gene expression related to osteoblasts [88] and alkaline phosphatase (ALP) of osteogenic-differentiating MSCs [89]. Marins et al. evaluated the effects of strontium ranelate on ligature-induced periodontitis in estrogen-deficient and estrogen-sufficient rats and found that strontium ranelate could prevent ligature-induced bone loss in an estrogen-deficiency situation and increase trabecular bone area in the presence and absence of periodontal collapse in states of estrogen deficiency and estrogen sufficiency to a certain extent. In addition, strontium ranelate also intervened the expression of bone markers, seeming to be anti-absorption [90]. Miranda et al. had evaluated the impact of strontium ranelate on the wound healing of tooth-extraction in estrogen-deficient and estrogen-sufficient rats, and found that strontium ranelate benefited bone healing and the expression of bone markers in estrogen-deficient rats, whereas its benefits in estrogen-sufficient rats were modest [91].

Ideal nano-DDSs for periodontal regeneration are expected to meet the requirement of simultaneously favoring the osseointegration with host tissues and inhibiting the activity of bacterial pathogens. As such, a strategy of incorporating ceramic-based nanoparticles into polymers has been proposed [92]. For example, a novel bone graft substitute named strontium- and zinc-containing bioactive glasses was synthesized and applied with various amounts of alginate, being capable of releasing strontium, zinc, and calcium ions in the Tris/HCl buffer to promote osteogenesis if tested in vivo. Another study has reported to load phenamil, an activator of bone morphogenetic protein (BMP), into Sr-doped mesoporous bioglass nanoparticles to stimulate the osteo/odontogenesis of human mesenchymal stem cells [93]. Zamani et al. have studied the positive impacts of incorporating zinc and magnesium ions into bioactive glasses composition and found that the incorporated composites could easily improve the antibacterial activity, mechanical properties, and bioactivity of alginate, showing great potential for bone tissue regeneration [94].

Silver and zinc-based nanoparticle (NP) are reported to play a significant role in inhibiting bacterial activity and promoting osteogenic properties [95,96], but the degradation of these NPs might lead to accumulation of heavy metal elements within the body. Recently, magnesium oxide NPs (nMgO) has gained considerable attention in biomedical applications of periodontal regeneration due to its advantages of antibacterial activity and osteoinductivity. Liu et al. have reported that poly(L-lactic acid) (PLA)/gelatin periodontal membrane containing MgO nanoparticles represented a dose-dependent magnesium ion-induced osteoinductivity of rabbit bone marrow stem cells (rBMSCs). It was inferred that the pH microenvironment beneficial for cell proliferation was ameliorated through neutralization of the alkaline ions of nMgO hydrolysis by the acidic degradation products of PLA [97]. Bilal et al. have also fabricated polycaprolactone (PCL)/gelatin core-shell nanocellulose periodontal membrane incorporating MgO nanoparticles, and found that the incorporation of MgO nanoparticles barely affected the mechanical properties and morphology of nanocellulose membranes and showed great potential for periodontal tissue regeneration [98].

The application of bone graft biomaterials in treatment of periodontal defects has achieved great success in a certain degree [99], but most of the applied bone graft biomaterials such as hydroxyapatite-HAp- and other calcium phosphates inevitably represent a relatively fast rate of biodegradation. Therefore, it is desirable to fabricate a biocompatible and non-resorbable composite for periodontal regeneration. Osorio et al. have covalently connected 2-hydroxyethyl methacrylate, ethylene glycol dimethacrylate, and methacrylic acid to fabricate zinc or calcium loaded PolymP-nActive polymeric nanoparticles (NPs), and found that the calcium-loaded nanoparticles were nontoxic and could promote precipitation of calcium phosphate deposits, which may provide a new strategy for treatment of periodontal defects [100].

#### 4.1.4. Dendrimers

Dendrimers are made up of unimolecular micelles with interior hydrophobic and exterior hydrophilic shells. The core can be composed of polymers as polypropylene imine (PPI), polyamidoamine (PAMAM), polyethylene glycol (PEG), and others, while the most used is PAMAM. The interior hydrophilic or hydrophobic shells allow dendrimers to encapsulate specific drug molecules; while the hyperbranched nanoparticle structures with peripheral functional groups also allow them to deliver specific drug molecules on the surface in multivalent manner (Figure 6) [101]. Therefore, dendrimers may serve as an optimal platform for efficient, sustained, controlled, and targeted drug delivery in tissue regeneration [102].

The PAMAM dendrimers are water-soluble molecules and can entrap hydrophobic molecules or other biomolecules to enhance aqueous solubility, dissolution, stability, and pharmacokinetics of various drugs; for instance, the cationic PAMAM binding anionic mucin displays well mucoadhesive properties to expand its use in oral cavity [103]. However, despite the numerous advantages, only a few studies have reported the application of dendrimers in treatment of periodontal defects. Gardiner et al. investigated PAMAM dendrimers for the delivery of the Triclosan (TCN), but observed that dendrimer could not maintain the increased solubility of TCN at lower pH. Then, a substituted method of directly integrating triclosan to PAMAM dendrimers via a hydrolysable linkage was proposed [104]. Lu et al. [105] and Backlund et al. [106] both researched the antibacterial activity of nitric oxide (NO)-releasing amphiphilic PAMAM dendrimers as NO can be used as an antibacterial agent for periodontal pathogens, such as *Aggregatibacter actinomycetemcomitans* and *Porphyromonas gingivalis*, which advocates the possible therapeutics for elimination of periodontitis.

**Table 2 pharmaceutics-14-02250-t002:** Summary on Nanoparticulate Delivery Systems for Periodontal Tissue Regeneration.

Nano-DDSs	Characteristic	Clinical Application	Research Findings	Reference
Liposomes	Structural versatility, biocompatibility, biodegradability, non-toxic and nonimmunogenicity	Bubble liposomes plus ultrasound	Providing an efficient technique for delivering plasmid DNA into the gingiva.	Sugano et al. [48,49]
		Negatively charged liposomes	A versatile tool in the field of drug-carrier systems due to their size and hydrophobic and hydrophilic character	Bozzuto et al. [50]
		Liposomes modified with viral fusion proteins	Exhibiting capabilities to fuse with or to disrupt endosomal and/or lysosomal membranes and introduce encapsulated antigenic into cell cytosol.	Kunisawa et al. [51]
		A novel pH-activated nanoparticle comprising a quaternary ammonium chitosan, i.e., N,N,N-trimethyl chitosan, a liposome, and doxycycline (TMC-Lip-DOX NPs)	Achieving superb inhibition of free mixed bacteria and biofilm formation, and showing excellent biocompatibility with human periodontal ligament fibroblasts	Hu et al. [52]
		Minocycline hydrochloride liposomes	Showing significantly stronger and longer inhibition of TNF-α secretion in macrophages compared to periocline and minocycline hydrochloride solution	Liu et al. [54]
		A therapeutic resveratrol-loaded liposomal system (Lipo-RSV)	A potential therapeutic system for the antibiotic-free treatment for periodontal diseases	Shi et al. [55]
		Stealth, long circulating or PEGylated liposomes	Increasing repulsive forces between liposomes and serum-components, reducing immunogenicity and macrophage uptake, enhancing the blood circulation half-life, and reducing the toxicity of encapsulated compound	Di et al. [57]
Polymeric Nanoparticles	Non-immunogenicity, biological inactivity, and the facility of functional groups for covalent coupling of drugs or target moieties	Chitosan (CHT)	An excipient for producing nanoparticles for the treatment of periodontal defects	Ul et al. [59]
		a combination of chitosan (CHT) with bioactive glass nanoparticles (BG-NPs)	Serving as a temporary guided tissue regeneration membrane in periodontal regeneration with the possibility to induce bone regeneration	Mota et al. [66]
		Chitosan/plasmid nanoparticles encapsulating platelet-derived growth factor (PDGF)	Offering a 3D carrier for increased proliferation of periodontal ligament cells	Peng et al. [67]
		Nanogels	Serving as suitable carriers for the delivery of a variety of chemotherapeutics, antisense nucleotides, siRNAs, and peptides	Hajebi et al. [68]
		Cholesterol-bearing pullulan (CHP)-nanogel	Working as a suitable carrier for the W9-peptide, preventing aggregation and increasing the stability of the W9-peptide	Alles et al. [73]
		Asymmetric barrier membranes based on polysaccharide micro-nanocomposite hydrogel	Showing better biocompatibility and higher mechanical properties, indicating its potential for periodontal tissue engineering	He et al. [74]
		Poly (lactic-co-glycolic acid) (PLGA)	Serving as a reference polymer in manufacturing of nanoparticles to encapsulate and deliver a wide variety of hydrophobic and hydrophilic molecules	Ortega-Oller et al. [75]
		A mixture of poly(lactic-co-glycolic acid)/chitosan/Ag nanoparticles	Having no cytotoxicity and contributed to cell mineralization	Xue et al. [76]
		Polytetrafluoroethylene (PTFE)	Being commonly used because of its porous microstructure that allows connective tissue in growth	Kameda et al. [77]
		Expanded polytetrafluoroethylene (e-PTFE)	Serving as a membrane barrier for regeneration procedures	Soldatos et al. [78]
		High-density polytetrafluoroethylene membranes (n-PTFE)	Being non-porous, dense, non-expanded and non-permeable	Carbonell et al. [79]
		Polycaprolactone (PCL)	Being capable of mimicking the extracellular matrix (ECM), combining both core-shell and nano-reservoirs functionalization	Bassi et al. [81]
		BMP-2 or BMP-2/Ibuprofen functionalized PCL membranes	Passive release of ibuprofen will decrease the inflammation leading to increased BMP-2 secretion by macrophages while active loading of BMP-2 or other growth factor will directly promote the regeneration of targeted tissue such as alveolar bone	Park et al. [83]
		Chorion membrane (CM) and amnion/chorion membrane (ACM)	Exerting the anti-inflammatory, antifibrotic, and antimutagenic properties and pain-relieving effects	Gulameabasse et al. [84]
Inorganic Nanoparticles and Nanocrystals	Chemical stability, thermal resistance, and long-lasting action	Strontium (Sr^2+^)/strontium ranelate	A cation that stimulates the differentiation of mesenchymal stem cells to develop into bone tissue by suppressing the activity of osteoclasts as bone resorption cells	Pilmane et al. [87]
		Mesoporous bioglass	Favoring the osseointegration with host tissues while inhibiting bacterial activity for better periodontal regeneration	Sriranganathan et al. [92]
		Silver and zinc-based nanoparticle	Exerting significant effects on inhibiting bacterial growth and promoting osteogenic properties	Gaviria et al. [95] & Yoo et al. [96]
		Magnesium oxide nanoparticle	Presenting superior antibacterial activity and osteoinductivity	Liu et al. [97] & Bilal et al. [98]
		Zinc or calcium loaded PolymP-nActive polymeric nanoparticles	Promoting precipitation of calcium phosphate deposits	Osorio et al. [100]
Dendrimers	Hyperbranched structures, multivalent and modifiable surface, interior hydrophilic or hydrophobic shells	Polyamidoamine (PAMAM)	Enhancing aqueous solubility, stability, dissolution, drug release, targeting and pharmacokinetics of various drugs	Chauhan et al. [101]
		PAMAM dendrimers solubilizing triclosan (TCN)	Failing to maintain the previous observations of increased solubility of TCN at lower pH	Gardiner et al. [104]
		PAMAM dendrimers and silica based nitric oxide (NO) release	Displaying considerably less toxicity for human gingival fibroblasts at the levels required to kill periodontal pathogens	Backlund et al. [106]

### 4.2. Nanofibers Delivery Systems

Nanofibers are another nanomedicine that shows promise for periodontal tissue regeneration due to their biomimetic behaviors, such as improving the cell adhesion, differentiation, proliferation and thus promoting the regeneration of the damaged periodontal tissue [107]. The advantageous characteristics of nanofibers over other forms of the same material are usually reflected in the high surface-to-volume ratio, theoretically unlimited length, ideal porous structure, better mechanical properties, and tunable flexibility [108]. Moreover, the option of incorporating nanoparticles into the nanofiber structures can provide a delivery of dual or more active ingredients for tissue regeneration [109]. For example, the nanofiber scaffolds incorporating silver and hydroxyapatite nanoparticles represent enhanced antibacterial activity, osteoinductivity, cellular viability and biodegradability [110].

Electrospinning is recognized as the most promising nanofiber production technique as it meets the application requirements of high processability, maneuverability, and desirable mechanical properties based on the selection of polymers [111]. In addition, along with biological and topographical signals, electrospun nanofiber membranes show great potential to offer an optimal microenvironment for cell proliferation and differentiation [112]. In addition, the electrospun nanofibers can be modified with biomolecules such as proteins and nucleotides to improve the local microenvironment for periodontal regeneration [113]. For instance, Shi et al. have designed an electrospun nanofiber-based membrane with infection-responsive character. The anti-infection drugs could be released in a targeted and efficient manner, and the polycaprolactone nanofiber membranes containing metronidazole exactly produced clear antibacterial zones around the GTR membranes [114].

Newer electrospinning methods show advantages over traditional types, whose product has become a multi-component fiber rather than single polymer. Coaxial electrospinning is one of the new technologies, which can produce “core-shell” fibers. Bioactive molecules make up the “core” while polymers make up the “shell” [115]. The capability of incorporating bioactive molecules into the core component allow core-shell nanofibers to provide a controlled drug delivery. Lam et al. have developed a multifunctional fibrous membrane that is also core-shell electrospun nanostructured. The membrane successfully controlled the delivery of enamel matrix derivatives (EMD) to PDLSCs [116]. The core-shell nanofiber membranes can concurrently act as mechanical barriers for infiltrative growth of epithelial cells and deliver bioactive molecules to promote tissue regeneration in periodontal defects [117]. In another study, Dos Santos et al. successfully prepared tetracycline hydrochloride (TH) loaded core-sheath nanofibers consisting of chitosan as the shell layer and poly(vinyl-alcohol) (PVA) as the core layer. The mechanical properties, stability, the release profile, and in vitro antimicrobial activity of this core-sheath nanofiber were all greatly improved, showing the potential of serving as a novel drug delivery system for periodontitis treatment [118]. In addition, neither natural nor synthetic polymers are perfect, but their blends combine to produce excellent mechanical and biological properties [119]. Among the numerous materials prepared for coaxial electrospinning, PCL is recognized to be most appropriate for electrospun cellulose nanofiber due to its mechanical properties, biocompatibility, and available fabrication. However, the applications of synthetic PCL are limited due to poor hydrophilicity and cell affinity. Gelatin is another most used natural polymer with cell-binding sites and biomolecular signature characters. The core-shell structured combination of PCL and gelatin can compensate the drawbacks of single PCL nanofibers [120]. In another study, Ranjbar-Mohammadi et al. blended poly(lactic glycolic acid) (PLGA), gum tragacanth (GT), and tetracycline hydrochloride (TCH) by electrospinning and coaxial electrospinning to fabricate a novel drug delivery system. Then the prolonged drug release, great antibacterial activity, biocompatibility, and other mechanical properties of the drug loaded core shell nanofibers showed great prospect for periodontal diseases [121].

Nanofibers can be fabricated and modified through various methods and the conjunction of specific ligands helps to promote the cellular activities such as proliferation and differentiation. Among the natural biopolymers as nanofibers drug carriers, silk fibroin (SF) is considered as one of the most used polymers due to its tailorable biodegradability, high biocompatibility, low bacterial adherence, high structural integrity, fabricability of customizable sizes, desirable mechanical properties, and interactions with biomolecular components. These characteristics, along with the advantageous benefits of electrospinning, allow SF to produce suitable nanofibers based on silk fibroin for drug delivery purposes [122]. However, the brittleness of SF restricts its use for flexible membrane material. Then, SF was mixed or reinforced with other materials, such as polyethylene oxide and chitosan, to achieve desired mechanical or biological properties [123]. In recent studies, a novel bilayer membrane comprising of nanocalcium-phosphate incorporating a silk fibroin-PCL-PEG-PCL electrospun layer and a PCL membrane layer was developed for GBR, and the results of structural, chemical and mechanical analyses prefigured the potential application for GBR treatments [124]. In addition, as SF nanofibers are not intrinsically osteoconductive, it is necessary to blend appropriate osteogenic factors to stimulate the activities of osteoblasts [125]. Shalumon et al. prepared nanofibrous SF/chitosan/nanohydroxyapatite/bone morphogenetic protein-2 (SF/CS/nHAp/BMP-2, SCHB2) composites to research the modulation of bone-specific tissue regeneration. In this nanofibrous membrane, BMP-2 was placed within the core while the shell was comprised of SF/CS/nHAp with two thicknesses (SCHB2-thick and SCHB-thin). Through the comparison of a controlled release profile, concentration of BMP-2 in the release media, release kinetics, osteoinductive activity toward human-bone-marrow-derived mesenchymal stem cells (hMSCs), and detection of gene expression between SCHB2-thick and SCHB-thin structures, nanofibrous SCHB2-thin mats were recognized as potential bone-tissue constructs [126]. In another study, BMP-2 loaded composites comprising of hydroxyapatite nanoparticles (HA) and silk fibroin (SF) showed great preservation of BMP-2 bioactivity and could provide tunable BMP-2 delivery to construct preferable microenvironments for bone regeneration [127]. Moreover, the character of possessing different amino acids allows SF polymers to provide numerous functional sites for biomineralization and promotes the modification of the nanofiber surface with various bioactive molecules to achieve the delivery of various growth factors [128].

The nanotechnology-based asymmetric membranes simultaneously provide a top layer that resists against physical damages and pathogen invasions, and an inner layer that allows inflammation elimination and bone regeneration. Then, Ghavimi et al. developed a nanofibrous asymmetric guided bone regeneration (GBR) membrane containing aspirin, which was reported to be able to control the balance between bone formation and resorption, accelerate bone repair, and prevent the differentiation and maturity of osteoclasts. The antibacterial activity, biocompatibility, osteoinductivity, and the release profile of aspirin were observed in this study. It was fortunate to find that the prepared membrane could keep soft tissue from infiltrating the bone defects, and the antibacterial activity and bone healing effects of this novel GBR membrane could promote bone regeneration [129].

## 5. Conclusions

The periodontal regenerative therapy is directed towards the reconstruction of lost tissues by forming new cementum, periodontal ligament and alveolar bone, and the therapeutic success is mainly dependent on antimicrobial and regenerative activity of the chemotherapeutic agents, polymers used for drug delivery, and targeted drug delivery systems. Given the ability to display material properties at the cellular and subcellular length-scales, nanotechnology has been adopted in regenerative medicine to embellish drug delivery systems that enable highly efficient local or systemic drug administration, prolonged retention in periodontal pockets, controlled and targeted release of precise medicines incorporated, etc. Emerging progresses in the field of nanotechnology offer multiple benefits and the development of modified nanomaterials is surely going to help to improve or restore the damaged periodontal tissues so long as the substances are nontoxic, biodegradable, and biocompatible. The exceptional features of nano-DDSs, such as tunable structures, customized surfaces, bioadhesive behavior, stimuli-responsive ability, and improved mechanical properties, allow for excellent periodontal tissue regeneration in animal models both in vitro and in vivo. Regardless, it is worth noting that the biosafety and nontoxicity of novel materials produced by nanotechnology is still an open question. The critical attention for the potential risks of nanoengineered drug delivery systems may constrain their clinical application, and thus lack credible proof in human body experiments. Therefore, there must be specific standardized methods to advance biomedical research and provide authentic and reliable proof of nano-DDSs for treatment of clinical periodontal defects. Fortunately, although the curative effect of nanotechnology for periodontal tissue regeneration is limited now, it is surely promising that nano-DDSs will progress rapid investigations to guarantee development and bring enormous therapeutical and economical potential for clinical applications in the future.

## Figures and Tables

**Figure 1 pharmaceutics-14-02250-f001:**
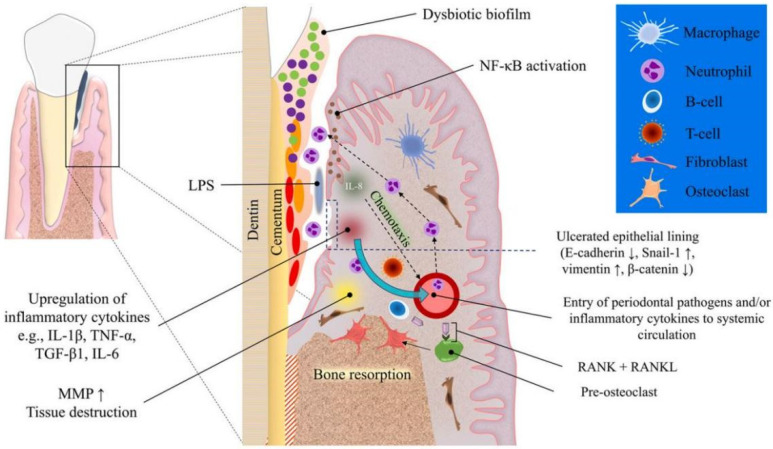
Schematic diagram of the pathogenesis of periodontitis. Dysbiosis of subgingival microbiome results in an increased populations of red and orange bacteria such as Porphyromonas gingivalis and Fusobacterium nucleatum. The virulence factors trigger an inflammatory response mainly via Toll-like receptor signaling which activates intracellular NF-κB which in turn increases interleukin (IL) production. IL-8, in particular, acts as a chemotactic agent for inflammatory cells such as neutrophils and macrophages, recruiting them from nearby blood vessels. Cellular (T-cells) and humoral (B-cells) immune cells also increase in numbers due to the chemotactic gradients generated in association with chronic inflammation. These inflammatory events lead to further escalation of inflammatory mediator production. The tissue destruction and bone resorption increase due to overexpression of matrix metalloproteinases and RANKL, respectively. Reprinted/adapted with permission from Ref. [11]. 2022, Saliem, S.S.

**Figure 2 pharmaceutics-14-02250-f002:**
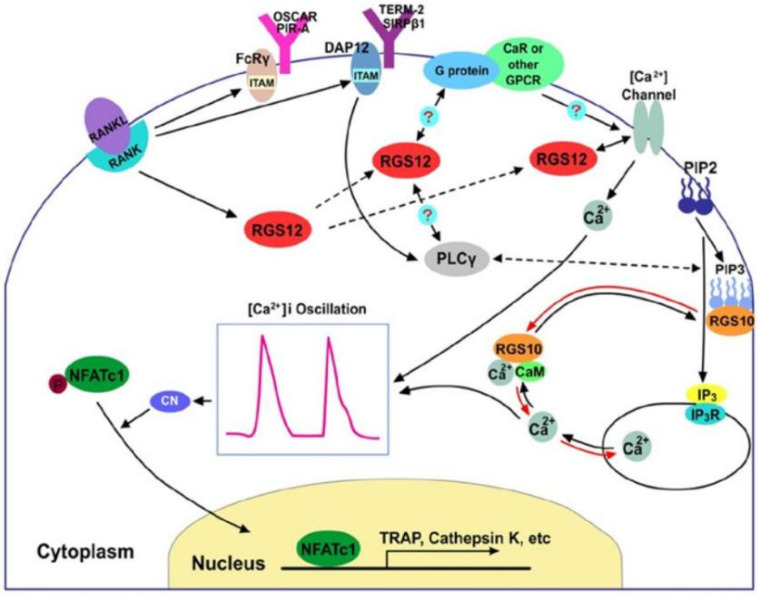
A proposed working model of RGS proteins in the regulation of the calcium oscillation-NFATc1 signal pathway for osteoclast differentiation. Reprinted/adapted with permission from Ref. [16]. 2014, Intini, G.

**Figure 3 pharmaceutics-14-02250-f003:**
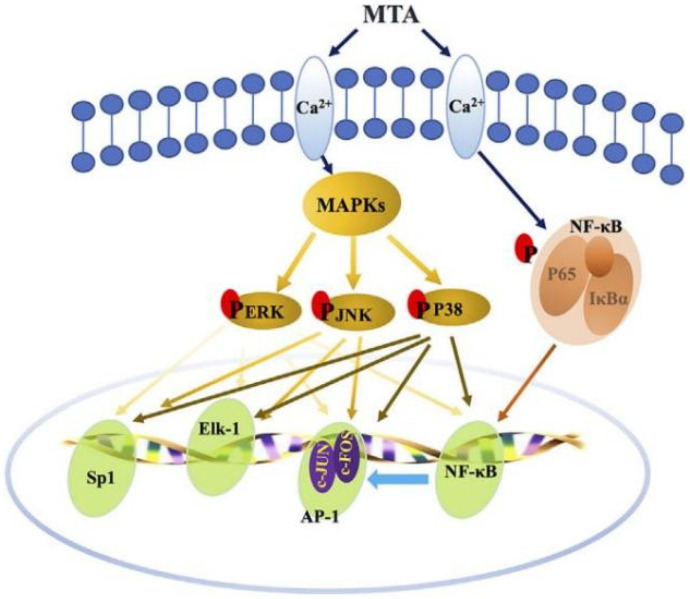
Schematic diagram of the activation of NF-κB and MAPK pathways by MTA treatment. Reprinted/adapted with permission from Ref. [23]. 2018, Yu, J.

**Figure 4 pharmaceutics-14-02250-f004:**
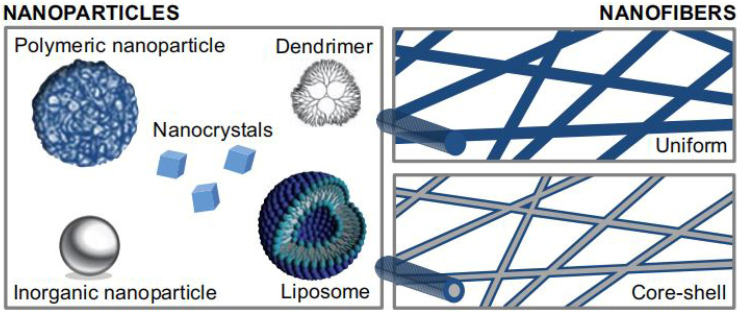
Various forms of nanomedicines. Reprinted/adapted with permission from Ref. [42]. 2015, Kristl, J.

**Figure 6 pharmaceutics-14-02250-f006:**
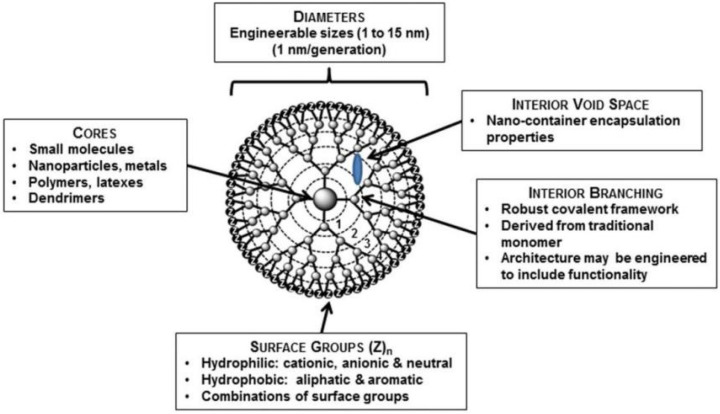
Schematic diagram of a G-4 dendrimer containing four generations (branching points), as indicated by generation numbers. Reprinted/adapted with permission from Ref. [101]. 2015, Chauhan, A.S.

## Data Availability

Not applicable.

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
