# Peer review of "Nano-Based Drug Delivery Systems for Periodontal Tissue Regeneration"

_pharmaceutics, 2022, doi:10.3390/pharmaceutics14102250_

Round 1

Reviewer 1 Report

The Review titled "Nano-Based Drug Delivery Systems for Periodontal Tissue  Regeneration" is well written and very interesting.

The aims are clearly described and well developed in the text.

The Review titled "Nano-Based Drug Delivery Systems for Periodontal Tissue  Regeneration" is well written and very interesting.

The aims are clearly described and well developed in the text.

My only request to the Authors is to change the term "thees" (page 10 line 331) to "these".

Moreover, the Authors could put the numerical coefficients as subscripts in the formulas.

Reviewer 2 Report

Review: Periodontitis often leads to progressive deterioration of ligament and alveolar bone resulting in gingival recession and functionally compromised dentition and includes critical challenges to reach efficacy and safety in controlling local inflammation, establishing regenerative. This draft provides an interesting overview of nano-based drug delivery systems and demonstrates their different applications.

1-More figures including different concepts should be included

2- An index or Scheme incorporating all sections should be helpful

3-Please include in conclusion future prospects

4-Include the state of the art of the clinical applications in this area.

5-Please include the following references related:

1-      Pharmaceutics 2022, 14, 1244. https://doi.org/10.3390/pharmaceutics14061244¸

2-      DOI: 10.3390/pharmaceutics14020455 and

3-      https://doi.org/10.1016/j.surfcoat.2020.126667

Reviewer 3 Report

A nice review article, well written and of interest not only for specialists.

Author Response

Thank you for your appreciation.